# Vancomycin Penetration in Brain Extracellular Fluid of Patients with Post-Surgical Central Nervous System Infections: An Exploratory Study

**DOI:** 10.3390/medicina61111989

**Published:** 2025-11-05

**Authors:** Skaistė Žukaitienė, Karolis Bareikis, Simona Stankevičiūtė, Akvilė Ūsaitė, Neringa Balčiūnienė, Tomas Tamošuitis, Romaldas Mačiulaitis

**Affiliations:** 1Institute of Physiology and Pharmacology, Medical Academy, Lithuanian University of Health Sciences, 44307 Kaunas, Lithuania; 2Kauno klinikos, Hospital of Lithuanian University of Health Sciences, 50161 Kaunas, Lithuania; 3Department of Neurosurgery, Medical Academy, Lithuanian University of Health Sciences, 50161 Kaunas, Lithuania; 4Department of Intensive Care, Medical Academy, Lithuanian University of Health Sciences, 50161 Kaunas, Lithuania

**Keywords:** vancomycin, microdialysis, brain extracellular fluid, post-surgical infection, pharmacokinetics, blood–brain barrier

## Abstract

*Background and Objectives*: Post-surgical central nervous system (CNS) infections are severe complications associated with high morbidity and mortality. Vancomycin is a key antibiotic used in their management. However, because of the restrictive properties of the blood–brain barrier (BBB), plasma concentrations may not accurately reflect drug exposure in the brain extracellular fluid (ECF), the presumed site of infection. Cerebral microdialysis enables direct measurement of unbound drug levels in brain ECF. This study aimed to assess vancomycin penetration into brain ECF in patients with suspected or confirmed post-surgical CNS infection. *Materials and Methods*: Five patients with suspected or confirmed post-surgical CNS infections were enrolled. Paired brain ECF microdialysate and plasma samples (and cerebrospinal fluid (CSF) samples, when available) were collected over two consecutive days at vancomycin steady state. Vancomycin concentrations were determined using a homogeneous enzyme immunoassay and corrected for probe recovery based on in vitro calibration. Pharmacokinetic parameters, including mean concentrations and 24-h area under the concentration–time curve (AUC24), were calculated for plasma and ECF, and ECF-to-plasma ratios were derived. *Results*: Two subgroups could be identified: patients with negligible ECF concentrations (“low penetrators”), and those with higher ECF levels (“high penetrators”). Mean (SD) ECF-to-plasma concentration ratios were 0.07 (0.04) in “low penetrators” and 0.44 (0.10) in “high penetrators”. The corresponding AUC24 ratios were 0.06 (0.03) and 0.40 (0.03), respectively. The presence of systemic inflammatory response syndrome (SIRS) was considered the most plausible factor differentiating these two subgroups. *Conclusions*: Vancomycin exposure in brain ECF demonstrated substantial interpatient variability in post-surgical CNS infections, with some patients showing minimal drug penetration.

## 1. Introduction

Post-surgical central nervous system (CNS) infections remain serious—though relatively rare—complications following neurosurgical procedures and are associated with high mortality, significant neurological sequelae, prolonged hospitalization, and increased healthcare costs [1,2,3,4,5,6,7]. These infections are typically healthcare-associated and caused by a broad spectrum of pathogens, including Gram-positive bacteria such as methicillin-resistant *Staphylococcus aureus* (MRSA) and coagulase-negative *staphylococci*, as well as multidrug-resistant (MDR) Gram-negative bacilli [8]. The reported incidence of post-surgical CNS infections varies substantially across studies—from about 1% to over 7% depending on patient characteristics and surgical type [9,10]—even if recent large-scale and meta-analytic analyses involving more than 30,000 patients indicate a lower overall rate of around 1.6–1.7% [11,12]. Reported risk factors for these infections include cerebrospinal fluid (CSF) leakage, perioperative corticosteroid use, external ventricular or lumbar drainage, repeat or prolonged surgery, increased intraoperative blood loss, malignancy, and a history of prior CNS infection [10,11,12].

Current guidelines recommend empiric combination therapy with vancomycin and an anti-pseudomonal β-lactam, tailored to local susceptibility data, and endorse vancomycin as the first-line treatment for MRSA and coagulase-negative staphylococcal infections unless the minimum inhibitory concentration (MIC) is ≥1 mg/L, in which case an alternative agent may be considered [8]. Vancomycin is a glycopeptide antibiotic that exerts its effect by binding to the D-alanyl-D-alanine terminus of peptidoglycan precursors, thereby blocking bacterial cell wall synthesis and ultimately causing cell death [13]. To maximize efficacy and minimize toxicity, vancomycin therapy should be guided by therapeutic drug monitoring (TDM) [14]. In critically ill patients, vancomycin trough concentrations of 15–20 mg/L are advised [8,14]. However, these recommendations are based largely on plasma concentrations and do not account for the restrictive properties of the blood–brain barrier (BBB), which limits vancomycin penetration into the brain [10]. For that reason, trough concentrations of to 20–25 mg/L are targeted in some practices [15,16]. However, there is currently no clear evidence linking higher plasma concentrations with improved efficacy or reduced relapse rates, indicating that these recommendations remain largely empirical. In contrast to sepsis, where area under the concentration—time curve (AUC)-based TDM is increasingly recommended, such an approach has not yet been established for CNS infections [14].

Although vancomycin is not formally authorized for CNS infections, it remains widely used “off-label” based on guideline recommendations, primarily due to the lack of equally effective alternatives [8,13,17]. In cases of vancomycin intolerance or elevated MICs, alternatives such as linezolid, daptomycin, or trimethoprim-sulfamethoxazole can be used, though clinical evidence for their efficacy in treating CNS infections remains limited [8]. Newer glycopeptides, including dalbavancin, telavancin, and oritavancin, have shown experimental activity, but their high protein binding and poor cerebrospinal fluid penetration restrict their clinical applicability [17]. Thus, despite these limitations, vancomycin remains the most reliable therapeutic option for MRSA-related post-surgical CNS infections worldwide.

Vancomycin is a large (1448 Da), hydrophilic compound that crosses the BBB mainly through passive paracellular diffusion. Its penetration into brain ECF is therefore largely determined by the structural integrity of the BBB [18,19]. Under physiological conditions, this barrier restricts the entry of molecules to those that are relatively small and lipophilic, generally below approximately 500 Da, which explains why vancomycin penetration into the CNS is limited [20,21,22]. However, in pathological states the BBB may become disrupted, allowing large hydrophilic molecules such as vancomycin to reach the brain [18,19]. CNS infection is a well-recognized factor that increases BBB permeability, and many other neurological conditions have also been associated with varying degrees of BBB disruption [23]. In our study population, both the underlying CNS disease and the surgical intervention were expected to contribute to some level of BBB compromise, leading us to hypothesize that this could facilitate vancomycin penetration into the brain.

Although numerous studies have investigated vancomycin concentrations in cerebrospinal fluid (CSF) [24,25,26,27], it remains uncertain whether CSF adequately reflects drug exposure at the actual site of infection—the brain extracellular fluid (ECF). Moreover, while CSF TDM has been applied to optimize target concentrations, particularly in cases receiving intrathecal vancomycin, current evidence indicates that no consistent associations have been demonstrated between CSF vancomycin levels and treatment efficacy or toxicity [27,28,29,30]. This uncertainty is further compounded by differences between the blood–brain and blood–CSF barriers [31,32], which limit the extrapolation of CSF data to brain tissue exposure. Microdialysis, a minimally invasive technique that enables frequent sampling of unbound drug concentrations directly from the brain ECF, offers a unique opportunity to overcome these limitations and to provide detailed pharmacokinetic (PK) and pharmacodynamic (PD) insights at the target site [33].

To date, data on vancomycin brain ECF exposure in patients with post-surgical CNS infections are extremely limited. Caricato et al. reported vancomycin concentrations in the brain ECF using cerebral microdialysis; however, this study included patients with severe traumatic brain injury and concomitant MRSA pneumonia rather than CNS infection [19], making direct extrapolation to the post-surgical meningitis population inappropriate.

This study aims to address this critical gap by using cerebral microdialysis to characterize vancomycin penetration into the brain ECF in patients with suspected or confirmed post-surgical CNS infections. By characterizing target-site pharmacokinetics, these findings may support rationalizing dosing strategies and ultimately improve clinical outcomes.

## 2. Materials and Methods

### 2.1. Study Design and Patients

This prospective observational microdialysis-based pharmacokinetic study, conducted at a tertiary hospital, aimed to evaluate vancomycin penetration into the brain ECF in patients with post-surgical CNS infections. All patients admitted to the neurointensive care unit with suspected or confirmed post-surgical CNS infection, who were receiving vancomycin either as monotherapy or as part of combination antibacterial therapy and undergoing routine multimodal neuromonitoring including microdialysis, were screened for eligibility. In our setting, multimodal neuromonitoring (including microdialysis) was routinely applied to all patients diagnosed with post-surgical CNS infection; therefore, no patients were excluded from the study due to absence of a microdialysis catheter. Microdialysis is considered the gold standard for determining drug concentrations at the target site of infection [33]. In this study, because patients were already undergoing microdialysis as part of routine multimodal neuromonitoring, no additional intervention was required, thereby preserving the purely observational nature of the study.

Patients were eligible if they were adults (≥18 years), had undergone neurosurgery for any indication, and subsequently developed bacterial CNS infection. Exclusion criteria included morbid obesity (body mass index (BMI ≥ 40 kg/m^2^), ongoing renal replacement therapy for acute or chronic renal failure, hepatic impairment classified as Child–Pugh class B or C, or severe illness reflected by an APACHE II score ≥ 25.

The study was conducted in accordance with the Declaration of Helsinki, and approved by the Kaunas Regional Biomedical Research Ethics Committee (protocol code BE-2-25; date of approval 5 May 2020). Eligible patients or, when necessary, their legally authorized representatives provided informed consent. Following consent, sample and data collection were initiated as described in the subsequent sections.

### 2.2. Treatment and Data Collection

The vancomycin (as vancomycin hydrochloride) used in this study was Vancomycin Dr. Eberth (Dr. Eberth Arzneimittel GmbH, Ursensollen, Germany). Initial loading and maintenance doses were prescribed based on individual patient characteristics, including body weight and renal function, and were subsequently adjusted according to measured plasma vancomycin concentrations.

Plasma and microdialysate sampling commenced once anticipated steady-state plasma concentrations had been reached. The time to steady state was calculated individually for each patient using standard pharmacokinetic equations (see Appendix A). To minimize error, microdialysis sampling was started at least one dosing interval after the calculated steady-state had been reached. Microdialysate samples were collected only after sufficient volume (approximately 60–70 µL) had accumulated in the collection vial, which typically required 4–6 h with the available microdialysis system. Each collected microdialysate sample represented the mean vancomycin concentration over the collection interval, rather than an instantaneous time-point concentration. Plasma samples were drawn at the same time when an adequate microdialysate volume was obtained to allow direct comparison.

To account for inter-day variability, sampling was performed on two consecutive days capturing at least two dosing intervals. In patients with external ventricular drainage, an additional CSF sample was collected once per study day. CSF concentrations reflected the average vancomycin exposure over the sampling interval, from the start of microdialysis collection to the end of each day.

Because the vancomycin concentration measured in the microdialysate represents only a fraction of the actual concentration in brain ECF, the values were corrected using an in vitro recovery calibration based on previously described methodology [34], resulting in the following equation for calculating brain ECF concentration:y = 0.313 + 1.141 × x
where x denotes the measured vancomycin concentration in the microdialysate.

To explore potential factors influencing BBB penetration of vancomycin, additional patient data were extracted from electronic health records. Collected variables included demographic and clinical parameters such as age, sex, height, weight, BMI, serum urea, creatinine, and creatinine clearance calculated using the original Cockcroft–Gault formula. Significant changes in renal function during the sampling period (defined as a change in creatinine clearance >20 mL/min, either improvement or deterioration) were also recorded, along with liver function parameters, serum albumin levels, and the type of vancomycin administration (intermittent versus continuous infusion). Information on intraventricular vancomycin administration (yes/no), CSF lactate levels, CSF/blood glucose ratio, cell index, and the presence of fever or other possible infection sites was documented, together with CSF culture results.

To relate ECF vancomycin levels to clinical efficacy, additional outcome data were collected. Short-term outcomes included resolution of fever, reduction in blood inflammatory markers, and overall clinical improvement as assessed by the treating physician. Long-term outcomes included infection cure or death attributable to central nervous system infection, duration of antibacterial therapy, length of stay in the neurointensive care unit, and total duration of hospitalization. The principal steps of the study design are illustrated schematically in Figure 1.

### 2.3. Vancomycin Administration

Vancomycin was administered either as intermittent infusions or as continuous infusion, at the discretion of the treating physician, and this variability was analyzed as a potential factor influencing penetration. Vancomycin powder was reconstituted with 20 mL of sterile water for injection to yield a stock solution of 50 mg/mL. For intermittent infusion, 1000 mg of the reconstituted solution was further diluted with at least 200 mL of 0.9% sodium chloride to achieve a final concentration not exceeding 5 mg/mL. The solution was administered intravenously at a maximum rate of 10 mg/min, with each infusion lasting at least 60 min. For continuous infusion, an initial loading dose was prepared and administered as described above, followed immediately by a continuous 24-h infusion. For this purpose, vancomycin was diluted in 0.9% sodium chloride and delivered as a 24-h intravenous drip infusion to provide the prescribed daily dose. At the discretion of the physician, some patients additionally received intraventricular vancomycin through external ventricular drains at a dose of 10–20 mg once daily for 5 days. For this purpose, the reconstituted vancomycin solution containing 500 mg vancomycin was diluted with at least 100 mL 0.9% sodium chloride, and the required dose (10–20 mg) was administered directly into the ventricles. TDM was performed in clinical practice according to institutional standards (target trough 15–20 mg/L), but dosing adjustments were not applied at the time of study sampling.

### 2.4. Microdialysis System

Microdialysis was performed using a 70 Microdialysis Bolt Catheter (M Dialysis AB, Stockholm, Sweden) with a polyurethane shaft measuring 130 mm in length and 0.9 mm in diameter. The semipermeable membrane, made of polyarylethersulfone (PAES), was 10 mm long and 0.6 mm in diameter, with a molecular weight cut-off of 20 kDa, allowing vancomycin diffusion while restricting larger molecules. The inlet and outlet tubing had internal diameters of 0.15 mm, with external diameters of 1.0 mm and 0.9 mm, respectively.

A 106 MD Pump (M Dialysis AB, Stockholm, Sweden) maintained a constant perfusion flow of 0.3 µL/min. During priming, a higher flow rate of 15 µL/min was applied for five minutes to ensure complete filling of the system, after which the pump automatically switched to the experimental flow rate. Dialysates were collected in 200 µL microvials (M Dialysis AB, Stockholm, Sweden). To prevent misinterpretation caused by the initially higher perfusion rate, the first microvial collected during the first hour was discarded.

### 2.5. Bioanalysis

Among the various analytical methods available for vancomycin determination, each with distinct advantages and limitations, our center employed the homogeneous enzyme immunoassay (Emit^®^ 2000 Vancomycin Assay, Siemens Healthcare Diagnostics Inc., Newark, DE, USA) for routine TDM at the time of the study [35], and therefore this method was chosen for the present investigation. The Emit^®^ 2000 Vancomycin Assay operates on a competitive inhibition immunoassay principle. Vancomycin in the sample competes with enzyme-labeled vancomycin (conjugated with glucose-6-phosphate dehydrogenase, G6PDH) for binding sites on specific antibodies. In the presence of vancomycin, less enzyme-labeled conjugate binds to the antibody, leading to increased enzyme activity. The active G6PDH enzyme catalyzes the conversion of oxidized nicotinamide adenine dinucleotide to its reduced form, causing a measurable change in absorbance. This change is detected spectrophotometrically and is directly proportional to the vancomycin concentration in the sample. All assays were conducted using a Beckman Coulter AU680 analyser (Beckman Coulter Inc., Brea, CA, USA), following the manufacturer’s protocol. Although the method is primarily validated for human plasma and serum, it was successfully adapted in this study for use with microdialysate (Perfusion Fluid CNS, M Dialysis AB, Stockholm, Sweden) and CSF. Calibration was performed using commercial vancomycin calibrators (Emit^®^ 2000 Vancomycin Calibrators, Siemens Healthcare Diagnostics Inc., Newark, DE, USA) within the range of 2–50 mg/L, according to the instrument operator’s manual, with multi-level quality controls included to ensure accuracy and reproducibility. Matrix effect evaluation with spiked perfusion fluid (microdialysates) and CSF confirmed recoveries within 85–115% of nominal concentrations, coefficients of variation below 15%, and calibration curve slopes comparable to serum-based calibrators. Based on these findings, no significant matrix effect was detected, and the assay was deemed appropriate for vancomycin quantification in microdialysate and CSF.

It should be noted that the enzyme immunoassay measures total vancomycin concentrations in plasma, while microdialysis selectively recovers only the unbound (free) fraction of vancomycin. To enable meaningful comparison between vancomycin concentrations in brain ECF and plasma, as well as to calculate the ECF-to-plasma AUC24 ratio, a conservative adjustment was applied based on literature-reported plasma protein binding values ranging from 30–55% [13].

### 2.6. Pharmakokinetic Assessments and Statistical Analysis

The primary PK parameter of interest was AUC24 in both plasma and brain ECF. AUC24 was calculated for each dosing interval.

For plasma samples, when intermittent vancomycin dosing was used and at least two post-distribution concentrations (1–2 h after infusion) were available, AUC24 was estimated using the trapezoidal equation-based Sawchuk–Zaske method, as previously described in the literature [36]. When more than two post-distribution concentrations were available, the sample closest to the first post-distribution “peak” and the sample closest to the pre-dose “trough” were selected for AUC24 calculation.

If only a single concentration was available within a dosing interval, a Bayesian approach using a population pharmacokinetic model for critically ill ICU patients [37] was applied, using the freely accessible ClinCalc vancomycin calculator (ClinCalc LLC, version 1.0, St. Louis, MO, USA) [38]. In cases where vancomycin was administered as a continuous infusion over 24 h, and more than one concentration was available, the trapezoidal method was used to calculate AUC24 by summing partial AUCs between each time point. No dedicated commercial PK software was used, as the study aimed to employ approaches readily transferable to routine clinical practice.

For brain ECF concentrations, because visual inspection of concentration–time profiles revealed no distinct peaks or troughs, the trapezoidal method used for continuous plasma infusions was applied.

When calculating brain ECF-to-plasma concentration ratios, we accounted for the fact that each microdialysate sample reflects the average concentration over the collection interval. Therefore, the corresponding plasma concentration was estimated as the mean of the two plasma values obtained at the beginning (*a*) and end (*b*) of the respective interval. In cases where vancomycin was administered as intermittent infusion, only post-distributional plasma concentrations were used for these calculations to avoid bias from early distributional peaks.

Descriptive statistics (mean, standard deviation, median, minimum, and maximum) were used. Correlation analyses were conducted to assess the relationship between plasma and brain ECF exposure (AUC24), as well as between individual plasma and brain ECF concentrations. All statistical analyses were performed using IBM SPSS Statistics version, 30.0.0.0 (172) (IBM Corp., Armonk, NY, USA).

## 3. Results

### 3.1. Patient Demographics, Neuroinfection Profile, and Clinical Outcomes

This study included five post-surgical patients (median age 53 years; range 28–74; four males, one female) with BMI ranging from 22.1 to 29.4 kg/m^2^ and variable renal function (creatinine clearance 82–271 mL/min). Key patient characteristics are summarized Table 1 below, while detailed individual characteristics, infection profiles, and clinical outcomes are provided in Appendix A).

All patients had preserved liver function and normal albumin levels. Neurosurgical conditions were heterogeneous, including congenital brain malformation (frontobasal encephalocele), traumatic brain injury with chronic subdural hematoma, hemorrhagic stroke with intracerebral and intraventricular hematoma, arteriovenous malformation complicated by secondary hemorrhage after embolization, and ruptured aneurysm with subarachnoid hemorrhage.

At the time neuroinfection was suspected, two patients were febrile, one was subfebrile, and two were afebrile. CSF analysis showed an elevated cell index (>5) in three patients, a CSF-to-blood glucose ratio < 0.4 in four patients (data missing for one), and CSF lactate > 3.5 mmol/L in four patients (one missing). Two patients had culture-confirmed infections caused by *Proteus mirabilis*, *Escherichia coli* and *Klebsiella pneumoniae*, or *Staphylococcus capitis*, whereas three remained culture-negative, and in one case the diagnosis of neuroinfection was ultimately excluded.

Vancomycin was administered intermittently in two patients and as continuous infusion in three, with a mean maintenance dose of 24.2 mg/kg/24 h (SD 9.5). Intraventricular vancomycin (10–20 mg daily) was used in two cases. Additional empiric antibiotics included ceftriaxone, ceftazidime, or meropenem.

The ICU stay for the index neuroinfection episode ranged from 4 to 8 days (median 5 days), while the total hospitalization ranged from 31 to 112 days (median 36 days). Vancomycin therapy duration varied between 4 and 39 days. Four patients were successfully cured, while one patient died from septic shock and multiple organ dysfunction which may have been caused by initial empiric therapy not covering a the pathogen later identified in a CSF culture.

### 3.2. TDM Results

In total, 76 samples from 5 different patients were obtained; thsese comprised 36 blood plasma samples; 32 microdialysate samples from brain ECF (out of 36 collected; four were excluded due to insufficient volume for analysis), and 8 CSF samples. These samples enabled the calculation of vancomycin AUC24 (mg·h/L) on 15 occasions, corresponding to 2 to 4 dosing intervals per patient. Additionally, these data allowed for comparisons of average vancomycin concentrations in brain ECF versus plasma across 15 paired occasions in five patients (ranging from 2 to 5 comparisons per patient). CSF concentrations were reported for reference only, without formal calculations or comparisons. Vancomycin concentrations in plasma and brain ECF were plotted over time on separate graphs to assess potential temporal trends. These time-concentration profiles are provided in Appendix A).

#### 3.2.1. AUC24 and Corresponding ECF/Plasma AUC Ratios

Total plasma AUC24 values demonstrated marked interindividual variability, ranging from 255 to 986 mg·h/L per patient and dosing interval (mean 612 mg·h/L, median 559 mg·h/L). Vancomycin exposure in ECF was consistently lower yet still showed considerable variation, ranging from 23 to 418 mg·h/L (mean 144 mg·h/L, median 134 mg·h/L) across different patients and dosing intervals. Two clear subgroups were distinguished based on vancomycin penetration: “low penetrators” with a mean (SD) AUC24 ECF-to-plasma unbound ratio of 0.06 (0.03), and “high penetrators” with a mean (SD) ratio of 0.40 (0.03). Notably, the classification of patients as “low” or “high penetrators” was made post hoc, following internal discussion, and was based on the presumed probability of achieving adequate unbound vancomycin concentrations in brain ECF for CNS infections caused by vancomycin-susceptible organisms. The detailed rationale for this categorization is provided in the Discussion section. The AUC24 ECF-to-plasma ratio (total) varied widely, from 0.04 to 0.59, with the poorest penetration observed in Patient 2 (0.04) and the highest in Patient 4 (0.46). When adjusted for estimated unbound plasma concentrations, the AUC24 ECF-to-plasma unbound ratios ranged from 0.06 to 0.84 (assuming 30% vancomycin protein binding) and 0.10 to 1.30 (assuming 55% protein binding). Notably, Patient 2 exhibited the lowest overall BBB penetration, whereas Patient 5 showed the highest, with AUC24 ECF-to-plasma unbound ratios occasionally exceeding 1.0. These findings are presented in detail in Table 2 and illustrated in Figure 2 for improved visualization.

A correlation analysis between plasma and brain ECF exposure was performed. There was a statistically significant, moderate positive correlation between plasma AUC_24_ and brain ECF AUC_24_, *r*(13) = 0.53, *p* = 0.043, indicating that plasma concentrations explained approximately 28% of the variability in brain ECF exposure.

#### 3.2.2. Vancomycin Concentrations and Corresponding ECF/Plasma Ratios

Similarly to AUC24 patterns, vancomycin concentrations in plasma and brain ECF showed substantial variability across patients and dosing intervals. Plasma total concentrations, averaged over defined post-dose time intervals, ranged from 10.7 to 37.4 mg/L, while corresponding brain ECF concentrations were consistently lower, averaging only 0.45 to 12.8 mg/L over the same intervals. When expressed as mean values per patient, ECF-to-plasma ratios varied markedly—from as low as 0.04 in Patient 2 to as high as 0.55 in Patient 5—allowing identification of two distinct subgroups: “low penetrators”, with a mean (SD) ECF-to-unbound plasma ratio of 0.07 (0.04), and “high penetrators”, with a mean (SD) ratio of 0.44 (0.10). After correcting for estimated unbound plasma fractions (assuming vancomycin protein binding of 30–55%), the calculated ratios occasionally exceeded 1.0, suggesting that during some post-dose intervals, average ECF exposure could be comparable to or even higher than unbound plasma levels. Notably, Patient 2 exhibited the poorest CNS penetration, whereas Patient 5 showed the highest, with ECF-to-unbound plasma ratios (assuming an unbound fraction of 0.45) reaching up to 1.56. These averaged post-dose interval findings are presented in detail in Table 3 and illustrated in Figure 3 for easier interpretation.

An exploratory correlation analysis between individual plasma and brain ECF concentrations showed a weak, non-significant positive association, *r*(13) = 0.30, *p* = 0.282, suggesting that plasma concentrations explained only about 9% of the variability in ECF concentrations.

#### 3.2.3. Vancomycin Concentration in CSF

Only two patients (Patients 1 and 4), both of whom received additional intraventricular vancomycin, demonstrated therapeutically relevant CSF concentrations (24.13 mg/L and 116 mg/L, respectively). In the remaining patients, CSF concentrations were either near or below the lower limit of quantification (LLOQ). For one patient, CSF concentration could not be obtained due to the absence of a ventricular drain (Figure 2).

## 4. Discussion

Vancomycin concentrations in brain ECF exhibited marked interindividual variability among patients with suspected or confirmed neuroinfection following neurosurgery. In some cases, ECF levels were relatively high, whereas in others, they remained minimal—suggesting substantial variability in BBB permeability to vancomycin. To the best of our knowledge, this is the first study that investigated vancomycin penetration into brain ECF in patients with suspected or confirmed post-surgical CNS infection. Unfortunately, the overall number of patients is small, and there is high variability of their clinical course and other baseline aspects, to make meaningful comparisons between those who achieved and have not achieved therapeutic concentrations in ECF.

These data raise concerns regarding the adequacy of standard vancomycin dosing regimens for achieving therapeutic exposure in the brain. In some patients—specifically Patients #1, #5 and possibly #4, are all classified as “high penetrators”—ECF exposures appeared sufficient to achieve a therapeutic effect, with mean ECF slightly exceeding the susceptibility breakpoint of 2–4 mg/L for *Staphylococcus aureus* and coagulase-negative staphylococci [39]. Notably, in Patient #1, this occurred only at plasma exposures above the recommended AUC24 range. However, this interpretation must be regarded as tentative. At present, no well-defined threshold exists for unbound vancomycin concentrations or AUC24 in brain tissue that reliably correlates with positive clinical outcomes. Current clinical guidelines are based on total plasma exposure metrics (e.g., AUC₍_24_₎/MIC 400–600 mg·h/L or trough concentrations of 15–20 mg/L for *Staphylococcus* infections) [14]. For CNS infections, some authors have suggested targeting higher total plasma concentrations (20–25 mg/L) to account for the known challenges of crossing the BBB; however, these recommendations are empirical and not supported by direct evidence regarding efficacy, safety, or target-site concentrations [15,16,24]. Therefore, extrapolating such plasma-based targets to brain ECF—where only the unbound fraction is measured—remains uncertain. In this analysis, we assumed that therapeutically relevant unbound concentrations in brain ECF would approximate those required in plasma. Considering plasma-based therapeutic recommendations, vancomycin protein binding, and *Staphylococcus* susceptibility breakpoints (2–4 mg/L), an unbound ECF concentration of ≥4–5 mg/L was considered indicative of potentially adequate exposure against susceptible pathogens. Additionally, patients were categorized based on ECF-to-plasma ratios, with values ≥ 0.4 interpreted as indicative of “high” penetration. This classification was determined by consensus among clinical pharmacologists, intensive care specialists, and neurosurgeons involved in the study, but it should be viewed as provisional and subject to revision as further data emerge—for instance, Patient 4 might be better characterized as an “intermediate penetrator”, with concentrations likely sufficient only for pathogens with MIC < 1 mg/L.

Interpretation of these findings is further complicated by the absence of microbiological confirmation in most patients. Without identification of the causative pathogens and their vancomycin susceptibility profiles, it is difficult to determine with certainty whether the measured ECF concentrations were clinically adequate or meaningful. Nevertheless, culture-negative bacterial CNS infections are relatively common in clinical practice (reported in 15–90% of postoperative cases [11,40]), which partly explains this aspect in our cohort. It should also be recognized that in the presence of vancomycin resistance, no concentration within the therapeutic range would be sufficient to achieve bacterial eradication. Moreover, in our limited cohort we observed cases with extremely poor penetration into brain ECF, which by itself may result in insufficient exposure irrespective of pathogen susceptibility, since even highly susceptible organisms cannot be eradicated if concentrations at the target site remain close to zero.

These findings raise an important clinical question regarding TDM. Plasma-based TDM, although widely implemented, may not always reflect the antibiotic exposure at the site of a CNS infection. Our findings support site-based monitoring, such as CSF or brain ECF sampling where feasible, to individualize vancomycin dosing strategies. While not practical for routine clinical use at present, such approaches could be valuable in research settings or in selected critically ill patients in whom achieving adequate CNS penetration is essential.

Outcome interpretation is further influenced by the clinical context (see Appendix A): Patient #1 was culture-negative and treated with both vancomycin and ceftriaxone; although cured, the effect of vancomycin cannot be isolated. Patient #4 had *Staphylococcus capitis* (MIC unavailable) isolated from CSF and was cured following vancomycin treatment. Patient #5 was later determined not to have a CNS infection, despite initially receiving vancomycin empirically. Given these heterogeneous clinical scenarios and the small cohort size, any conclusions regarding the adequacy of vancomycin CNS exposure should be drawn with caution.

Beyond the uncertainty surrounding concentration adequacy, what clearly emerges from these observations is the marked variability in vancomycin penetration into brain ECF—likely reflecting differences in BBB integrity and patient-specific pathophysiological factors. Nearly all neurological diseases, including stroke, intracerebral hemorrhage, and traumatic brain injury, have been associated with some degree of BBB impairment [23]. Moreover, several systemic factors—such as level of meninges inflammation, aging, sex-related differences, fever, obesity, time, hypoxia, and gut microbiota dysregulation—have also been linked to BBB dysfunction—either decreasing BBB permeability, increasing or both at the same time [20,21,22,41,42]. These factors may collectively contribute to the observed variability in vancomycin penetration into the brain.

Assessing BBB disruption in clinical practice remains challenging. Several biomarkers have been investigated, e.g., white blood cells or Total Cells Ratio, serum ubiquitin carboxy-terminal hydrolase L1, transforming growth factor β, plasma and CSF norepinephrine levels, the CSF/serum immunoglobulin G ratio, platelet-derived growth factor receptor β, angiopoietin 2, YKL 40, and a number of others [25,43,44,45,46]. Among these, the CSF/serum albumin quotient (QAlb) is the most widely recognized and clinically accessible marker of BBB dysfunction. BBB breakdown may also manifest directly or indirectly through vasogenic edema, elevated intracranial pressure, and progressive neurological deterioration, which can serve as surrogate indicators of barrier impairment [47,48].

Considering the multiple factors potentially influencing BBB disruption, we attempted to explore why vancomycin concentrations in brain ECF varied considerably among our patients. Three factors were found to differentiate the “high penetrator” subgroup from the “low penetrator” subgroup: presence of fever, renal hyperfiltration (RHF), and, to some extent, intraventricular vancomycin administration. All other parameters that could potentially compromise BBB function, including perifocal edema, severe neurological impairment (Glasgow Coma Scale ≤ 8), comparable BMI, underlying neurological disease, and type of vancomycin infusion, were observed in both groups.

All patients in the high-exposure group were febrile or subfebrile, whereas both patients in the low-exposure group were afebrile. Additionally, two of the three “high penetrators” received intraventricular vancomycin, while none of the “low penetrators” did.

RHF emerged as the most biologically plausible and consistent differentiating factor. Although this association might initially appear counterintuitive, RHF has been observed in up to 50–85% of patients with conditions such as sepsis, subarachnoid hemorrhage, traumatic brain injury, or CNS infections—all of which can provoke systemic inflammatory response syndrome (SIRS). SIRS-related inflammatory cascades can lead to increased cardiac output and capillary permeability, alongside reduced systemic vascular resistance, collectively enhancing renal perfusion and glomerular filtration rate in patients with preserved kidney function [49].

In our cohort, all three patients classified as “high penetrators” exhibited more extensive CNS damage, which may have triggered a more pronounced systemic inflammatory response and thereby contributed to both RHF and increased BBB permeability. Although an attempt was made to compare commonly available inflammatory markers (e.g., C-reactive protein, absolute neutrophil count) across groups, no meaningful differences were observed—likely due to their limited specificity in this context.

To the best of our knowledge, only one previous study has assessed vancomycin penetration into brain ECF. This study included four patients who underwent craniotomy for the evacuation of cerebral post-traumatic hemorrhage with concomitant pneumonia. The maximum cerebral ECF concentration of vancomycin was 1.2 mg/L, classifying all these patients as “low penetrators,” similar to two patients in our study [19]. Notably, no adjustments for recovery (neither in vitro nor in vivo) were performed, so true concentrations might have been higher. The authors suggested that the low concentrations could be due to insufficient BBB disruption and concluded that in edematous brain tissue, vancomycin levels do not differ from those in healthy subjects. This finding contrasts with our results, as patient #5—who was later determined not to have a bacterial neuroinfection and was otherwise comparable to the patients in the study by Caricato et al. [19]—had one of the highest vancomycin exposures in brain ECF. This underscores the complexity of vancomycin penetration into brain ECF and suggests that multiple factors may contribute to achieving sufficient brain concentrations.

## 5. Limitations and Future Directions

This study has several limitations. The first and the most important is that we adjusted vancomycin levels in microdialysate from brain ECF using in vitro recovery data. This approach likely underestimates true concentrations, as in vitro recovery does not account for brain tissue properties. As demonstrated by Shroepf et al. [50], in vivo recovery is consistently lower; thus, the true concentrations in our patients’ brain ECF might be approximately one-third higher than reported. Consequently, concentrations in brain ECF in our study should be interpreted as “not lower than” rather than as absolute values.

Secondly, the variability in brain ECF concentrations may have been influenced by additional parameters, such as QAlb, a known marker of BBB disruption. However, QAlb is not routinely measured in our clinical setting and, given the observational design of our study, additional testing outside standard clinical practice was not feasible. As a result, the relationship between QAlb and vancomycin concentrations in brain ECF could not be evaluated.

Finally, our study included only a small number of patients, which limited the use of inferential statistics. Sampling was further constrained by low microdialysis flow rates and the relatively large sample volumes required for bioanalysis, reducing the ability to fully characterize vancomycin pharmacokinetics in the brain. Moreover, because each microdialysate sample reflects average concentrations over 4–6 h intervals, temporal resolution was limited and short-term pharmacokinetic fluctuations, such as immediate post-infusion peaks, could not be captured.

Future research should aim to more thoroughly explore the impact of SIRS on vancomycin penetration into the CNS, as well as to identify additional patient- or disease-related factors that may influence drug distribution into brain ECF. Further efforts are also needed to comprehensively characterize the pharmacokinetic profile of vancomycin within the CNS. Ideally, future studies should include a more homogeneous and targeted patient population with microbiologically confirmed post-surgical bacterial brain infections caused by vancomycin-susceptible microorganisms. This would enable investigation of the relationship between unbound vancomycin concentrations at the site of infection and outcomes in terms of bacterial eradication and clinical response. In addition, comparative evaluation of vancomycin with other antimicrobials used in the treatment of post-surgical CNS infections may help contextualize its pharmacological limitations and define situations where alternative or adjunctive therapies could be more appropriate. Multicenter prospective studies are warranted to overcome small-sample limitations and improve generalizability. In addition, development of predictive pharmacokinetic models, incorporation of pharmacogenomic and biomarker analyses, and exploration of advanced bioanalytical methods with lower sample volume requirements (e.g., <10 µL microdialysis) could further refine individualized therapy. Future work should also evaluate interventional strategies, such as higher systemic dosing or combined systemic and intraventricular administration.

## 6. Conclusions

Vancomycin exposure in brain extracellular fluid after post-surgical CNS infection showed marked inter-subject variability, with some patients achieving potentially therapeutic levels and others minimal penetration. These exploratory findings highlight the risk of subtherapeutic exposure despite adequate plasma concentrations and warrant confirmation in larger, controlled studies.

## Figures and Tables

**Figure 1 medicina-61-01989-f001:**
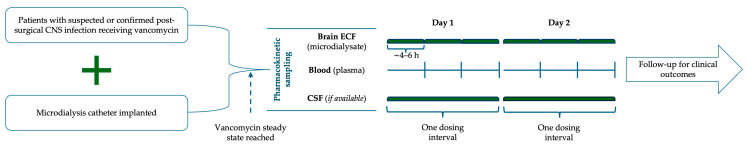
Schematic study design. Patients with suspected or confirmed post-surgical CNS infection receiving vancomycin and implanted with a microdialysis catheter underwent pharmacokinetic sampling after reaching steady state. Brain ECF (microdialysate), plasma, and CSF (if available) were collected over one dosing interval on Days 1 and 2, followed by clinical outcome monitoring. CSF—cerebrospinal fluid; ECF—extracellular fluid.

**Figure 2 medicina-61-01989-f002:**
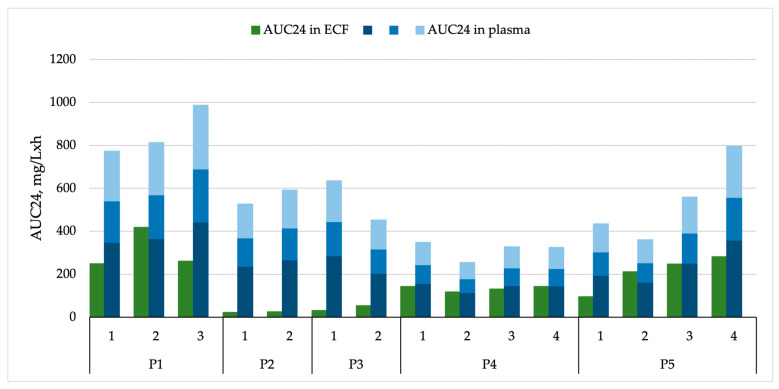
Vancomycin AUC24 in brain ECF and plasma. Different intensities of blue represent total plasma concentrations (light blue), unbound plasma concentrations assuming 30% protein binding (medium blue), and unbound plasma concentrations assuming 55% protein binding (dark blue).

**Figure 3 medicina-61-01989-f003:**
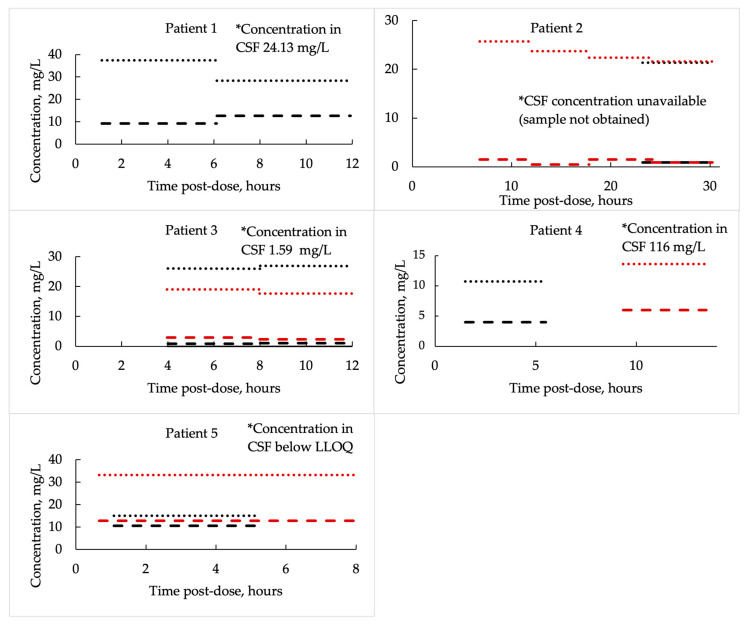
Vancomycin concentrations in plasma and brain extracellular fluid, averaged over defined post-dose time intervals. Round dots indicate plasma concentrations, while dashed lines represent brain ECF levels. Different colors correspond to separate dosing intervals.

**Table 1 medicina-61-01989-t001:** Key patient characteristics. Demographic and clinical data of enrolled patients, including age, sex, renal function (creatinine clearance and dynamic changes), neurological status (GCS score), and underlying surgical condition at study inclusion.

Patient No.	Age (Years)	Sex	ClCr (mL/min)	Renal Function Change	GCS Score	Underlying Surgical Condition
1	28	Male	146	No	6–7	Frontobasal encephalocele
2	69	Male	92	Yes (decrease)	10	Traumatic brain injury with subdural hemorrhage
3	74	Male	82	No	5	Hemorrhagic stroke: intracerebral & intraventricular hematoma
4	40	Female	271	No	7	Arteriovenous malformation with secondary intracerebral & intraventricular hemorrhage after embolization
5	53	Male	183	No	10	Ruptured aneurysm with subarachnoid hemorrhage

ClCr—creatinine clearance; GCS—Glasgow Coma Scale.

**Table 2 medicina-61-01989-t002:** Area under the concentration–time curve over 24 h (AUC24, mg·h/L) and AUC24 brain extracellular fluid/plasma ratio (AUC24_ECF_/AUC24_Plasma_).

Patient	Dosing Interval	AUC24 (mg·h/L)				AUC24_ECF_/AUC24_Plasma_ Ratio		
		Plasma			Brain ECF	Total	* Unbound 0.7 * *	* Unbound 0.45 * *
		Total	* Unbound 0.7 * *	* Unbound 0.45 * *				
* 1	1	773	* 541 *	* 348 *	249	0.32	* 0.46 *	* 0.72 *
	2	813	* 569 *	* 366 *	418	0.51	* 0.73 *	* 1.14 *
	3	986	* 690 *	* 444 *	261	0.26	* 0.38 *	* 0.59 *
	Mean (SD)	857 (113)	* 600 (79) *	* 386 (51) *	309 (94)	0.37 (0.13)	* 0.52 (0.19) *	* 0.82 (0.29) *
2	1	526	* 368 *	* 237 *	23	0.04	* 0.06 *	* 0.10 *
	2	592	* 414 *	* 266 *	26	0.04	* 0.06 *	* 0.10 *
	Mean (SD)	559 (47)	* 391 (33) *	* 252 (21) *	24 (2)	0.04 (0.00)	* 0.06 (0.00) *	* 0.10 (0.00) *
3	1	635	* 444 *	* 286 *	32	0.05	* 0.07 *	* 0.11 *
	2	451	* 316 *	* 203 *	53	0.12	* 0.17 *	* 0.26 *
	Mean (SD)	543 (130)	* 380 (91) *	* 244 (58) *	43 (15)	0.08 (0.05)	* 0.12 (0.07) *	* 0.19 (0.11) *
4	1	348	* 243 *	* 157 *	143	0.41	* 0.59 *	* 0.92 *
	2	255	* 178 *	* 115 *	117	0.46	* 0.66 *	* 1.02 *
	3	327	* 229 *	* 147 *	131	0.40	* 0.57 *	* 0.89 *
	4	323	* 226 *	* 146 *	143	0.44	* 0.63 *	* 0.98 *
	Mean (SD)	313 (40)	* 219 (28) *	* 141 (18) *	134 (12)	0.43 (0.03)	* 0.61 (0.04) *	* 0.95 (0.06) *
5	1	434	* 304 *	* 195 *	96	0.22	* 0.32 *	* 0.49 *
	2	361	* 252 *	* 162 *	211	0.59	* 0.84 *	* 1.30 *
	3	559	* 391 *	* 252 *	247	0.44	* 0.63 *	* 0.98 *
	4	796	* 557 *	* 358 *	281	0.35	* 0.50 *	* 0.79 *
	Mean (SD)	537 (191)	* 376 (134) *	* 242 (86) *	209 (81)	0.40 (0.22)	* 0.57 (0.22) *	* 0.89 (0.34) *

* Italic and gray text indicate assumed, not directly measured, values. Plasma unbound (0.7–0.45) corresponds to estimated unbound plasma exposure, calculated based on literature-reported vancomycin protein binding of 30–55%. AUC24: 24-h area under the curve; ECF—extracellular fluid.

**Table 3 medicina-61-01989-t003:** Vancomycin concentrations in plasma and brain ECF, averaged over defined post-dose time intervals and corresponding ECF/Plasma ratios.

Patient	Post-Dose Time (h)	Concentration, mg/L				ECF/Plasma Ratio	* ECF/Plasma Ratio (0.7) * *	* ECF/Plasma Ratio (0.45) * *
		Plasma Total	* Plasma Unbound (0.7) * *	* Plasma Unbound (0.45) * *	ECF			
1	1–6	37.39	* 26.17 *	* 16.83 *	9.24	0.25	* 0.35 *	* 0.50 *
	6–12	28.33	* 19.83 *	* 12.75 *	12.71	0.45	* 0.64 *	* 1.00 *
Mean (SD)		32.86 (6.40)	* 23.00 (4.48) *	* 14.79 (2.88) *	10.98 (2.45)	0.35 (0.14)	* 0.50 (0.20) *	* 0.77 (0.32) *
2	23–30	21.38	* 14.96 *	* 9.62 *	0.86	0.04	* 0.06 *	* 0.09 *
	7–12	25.72	* 18.00 *	* 11.57 *	1.48	0.06	* 0.08 *	* 0.13 *
	12–18	23.70	* 16.59 *	* 10.67 *	0.45	0.02	* 0.03 *	* 0.04 *
	18–24	22.39	* 15.67 *	* 10.08 *	1.48	0.07	* 0.09 *	* 0.15 *
	24–30	21.59	* 15.11 *	* 9.72 *	0.86	0.04	* 0.06 *	* 0.09 *
Mean (SD)		22.96 (1.79)	* 16.07 (1.26) *	* 10.33 (0.81) *	1.03 (0.48)	0.04 (0.02)	* 0.06 (0.03) *	* 0.10 (0.04) *
3	4–8	26.01	* 18.21 *	* 11.71 *	0.86	0.03	* 0.05 *	* 0.07 *
	8–12	26.88	* 18.82 *	* 12.10 *	1.07	0.04	* 0.06 *	* 0.09 *
	4–8	18.99	* 13.29 *	* 8.54 *	2.91	0.15	* 0.22 *	* 0.34 *
	8–12	17.61	* 12.33 *	* 7.92 *	2.29	0.13	* 0.19 *	* 0.29 *
Mean (SD)		22.37 (4.75)	* 15.66 (3.33) *	* 10.07 (2.14) *	1.78 (1.07)	0.09 (0.06)	* 0.13 (0.09) *	* 0.20 (0.14) *
4	2–6	10.72	* 7.51 *	* 4.83 *	3.99	0.37	* 0.53 *	* 0.83 *
	9–13	13.62	* 9.54 *	* 6.13 *	5.97	0.44	* 0.63 *	* 0.97 *
Mean (SD)		12.17 (2.05)	* 8.52 (1.43) *	* 5.48 (0.92) *	4.98 (1.40)	0.41 (0.05)	* 0.58 (0.07) *	* 0.90 (0.10) *
5	1–5	15.02	* 10.52 *	* 6.76 *	10.57	0.70	* 1.01 *	* 1.56 *
	1–8	33.15	* 23.21 *	* 14.92 *	12.84	0.39	* 0.55 *	* 0.86 *
Mean (SD)		24.09 (12.82)	* 16.86 (8.97) *	* 10.84 (5.77) *	11.71 (1.61)	0.55 (0.22)	* 0.78 (0.32) *	* 1.21 (0.50) *

* Italic and gray text indicate assumed, not directly measured, values. Plasma unbound (0.7–0.45) corresponds to estimated unbound plasma levels, calculated based on literature-reported vancomycin protein binding of 30–55%. Time intervals are calculated from the start of each infusion (intermittent) or from the start of continuous infusion; duplicate nominal intervals reflect different dosing days. SD: standard deviation; ECF: extracellular fluid.

## Data Availability

The original contributions presented in this study are included in the article/Appendix A. Further inquiries can be directed to the corresponding author(s).

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
