# Peer review of "Vancomycin Penetration in Brain Extracellular Fluid of Patients with Post-Surgical Central Nervous System Infections: An Exploratory Study"

_medicina, 2025, doi:10.3390/medicina61111989_

Round 1

Reviewer 1 Report

Comments and Suggestions for Authors

Reviewer Report

General Assessment

The manuscript addresses an important and clinically relevant problem: vancomycin penetration into brain extracellular fluid (ECF) in post-surgical bacterial CNS infections. The use of microdialysis, plasma, and CSF monitoring provides a unique and valuable dataset. However, certain sections of the manuscript require clarification, restructuring, and deeper interpretation to strengthen scientific rigor and clinical impact. Below are section-wise comments.

Title of study:

The title of the study should be modified as there is a clear mistake in title:

Vancomycin Penetration in Brain Extracellular Fluid of Patients with Post-Surgical Central Nervous System Infections: An Exploratory Study

This exploratory study is novel in applying microdialysis to measure vancomycin levels in brain ECF of post-surgical CNS infection patients, revealing clinically meaningful variability in drug penetration. Despite its limitations (small cohort, technical constraints, and lack of outcome correlation), it provides a valuable starting point for personalized antibiotic strategies and justifies larger, mechanistic studies to optimize therapy. The article has following key findings:

First-of-its-kind in post-surgical CNS infections using microdialysis to directly quantify vancomycin in brain extracellular fluid.

Demonstrates distinct pharmacokinetic subgroups (“low” vs. “high” penetrators), a concept not previously defined in this patient population.

 Highlights systemic inflammatory response syndrome (SIRS) as a potential modulator of vancomycin brain penetration, introducing a clinically relevant biomarker for therapeutic optimization.

But the study lacks some key points:

 Small sample size (n=5): The limited number of patients reduces statistical power and may not capture the full range of interpatient variability.

Single-center design: Findings may not be generalizable to broader or more diverse clinical populations.

Short observation period: Sampling was limited to two days at steady state, which may not reflect long-term pharmacokinetic fluctuations.

Lack of standardized clinical correlation: Therapeutic success (clinical or microbiological cure) was not directly correlated with measured ECF concentrations.

Potential technical variability in microdialysis: In vitro calibration may not fully correct for in vivo probe recovery differences.

Absence of control groups:  No comparison with non-surgical or non-infected patients to assess baseline penetration patterns.

Abstract

Abstract is too long and has casual language style; the author could make it more objective and understandable with concise findings and scope, therefore revise/modify the whole abstract with clear and accurate precise text and grammatically sound sentences.

INTRODUCTION:

PLEASE give a sound background of Vancomycin and mechanism of action its choice for treatment and its efficacy against MRSA with latest references. Actually this is the mist of our study that why we are going to study this drug/antibiotic for nervous system infections? Is there any more effective drug is available with better results than vamcomycin? So please make a clear picture of this antibiotic for its application in this study and globally with latest references.

On line 60: AUC has been used for the first time with its full version so make this abbreviation Area under the concentration–time

  1. Methodology:

Although no randomization or control group, so causality (e.g., between SIRS and higher vancomycin penetration) cannot be firmly established. Please justify?

The author described that only those patients already undergoing multimodal neuromonitoring including microdialysis were included. This introduces selection bias and may not represent the wider population of post-surgical CNS infection patients. How the author is going to balance it I this study?

How the author could logically justify that findings may be influenced by local patient demographics, surgical practices, or infection patterns, reducing external validity?

The study design focuses on pharmacokinetics but does not correlate ECF concentrations with clinical or microbiological outcomes (e.g., cure, relapse, mortality). Please justify.

The sampling only at steady-state over two consecutive days may miss variability during early therapy or under fluctuating clinical conditions. Please elaborate.

The author should also calrify the following points:

Dose Administration and Adjustment

Inconsistent dosing strategy: Vancomycin was given either intermittently or as continuous infusion at the discretion of the treating physician. This introduces variability and makes comparisons between patients unreliable.

Lack of standardized therapeutic target: Although plasma concentrations were monitored, the protocol does not clearly define target trough/AUC levels for dose adjustment.

The author should address the following technical questions:

Why was vancomycin dosing not standardized (intermittent vs. continuous infusion)?

What therapeutic targets (trough/AUC) were used for dose adjustment?

Does the 4–6 hour delay in microdialysate collection affect temporal resolution?

How reliable is simultaneous plasma–dialysate sampling when based only on dialysate volume availability?

Why was only in vitro calibration used instead of in vivo probe recovery methods?

How was “steady-state” plasma concentration objectively determined?

Can the correction equation from in vitro calibration be validly applied in vivo?

How was patient heterogeneity (renal function, comorbidities, CNS pathology) controlled?

Is one CSF sample per day sufficient to describe CSF pharmacokinetics?

How were short-term outcomes standardized beyond subjective physician assessment?

Could fever or biomarker changes reflect factors other than vancomycin penetration?

Why were only >20 mL/min changes in creatinine clearance considered significant?

Why was AUC/MIC not calculated despite plasma concentrations being measured?

  1. Results:

I have some technical questions in results to be addressed by author:

  1. How does the wide renal function range (66–271 mL/min) impact vancomycin PK/PD outcomes?
  2. Could heterogeneity in neurosurgical conditions confound vancomycin penetration results?
  3. What is the significance of culture-negative infections in interpreting vancomycin efficacy?
  4. Why was vancomycin used despite pathogens including Gram-negative bacteria (Proteus, E. coli, Klebsiella)?
  5. Did intraventricular vancomycin significantly alter brain ECF levels compared to IV-only patients?
  6. How did intermittent versus continuous infusion regimens influence brain penetration ratios?
  7. Is one patient death attributable to inadequate vancomycin penetration or inappropriate empiric coverage?
  8. Do elevated CSF lactate levels and reduced CSF/blood glucose ratios correlate with vancomycin penetration?
  9. How does short ICU stay (4–8 days) align with relatively long vancomycin therapy (up to 39 days)?
  10. Could variability in hospital stay (31–112 days) reflect differences in drug exposure or underlying pathology?
  11. Why was treatment success achieved in culture-negative cases despite uncertain pathogen susceptibility?
  12. What factors could explain the extreme interindividual variability in plasma AUC24 (255–986 mg·h/L)?
  13. Why is vancomycin ECF exposure consistently lower than plasma levels despite high systemic doses?
  14. What clinical or biological factors separate “low penetrators” from “high penetrators”?
  15. Is the observed AUC24 ECF-to-plasma ratio range (0.04–0.59) clinically sufficient to ensure therapeutic efficacy in CNS infections?
  16. Why did Patient 2 demonstrate such poor penetration (AUC ratio 0.04) compared to Patient 4 (0.46)?
  17. How should cases with unbound AUC ratios exceeding 1.0 (e.g., Patient 5) be interpreted—true drug accumulation in ECF or methodological artifact?
  18. Could differences in protein binding assumptions (30% vs 55%) significantly alter the clinical interpretation of CNS drug exposure?
  19. Do patients classified as “high penetrators” have better clinical outcomes compared to “low penetrators”?
  20. How do these findings challenge the practice of plasma-based vancomycin therapeutic drug monitoring (TDM) for CNS infections?

In Table 1:

Patient-specific variability

  1. Why did Patient 2 show consistently poor brain penetration (mean total ratio 0.04), while Patient 4 showed the highest penetration (mean total ratio 0.43)?
  2. What clinical or biological differences explain Patient 5’s wide range of ratios (0.22–0.59 total; up to 1.30 unbound 0.45)?
  3. Why did Patient 1 have ratios >1.0 (unbound 0.45) in some intervals, suggesting higher ECF exposure than plasma?

Methodological considerations

  1. How reliable are unbound plasma estimates based on assumed protein binding (30–55%) rather than direct measurement?
  2. Could differences in sampling intervals (e.g., 2 vs. 4 intervals per patient) bias mean AUC24 values?
  3. What is the significance of high intra-patient variability across dosing intervals (e.g., Patient 5, AUC24 plasma range 361–796 mg·h/L)?

Pharmacokinetic implications

  1. Why do high penetrators (Patients 4 and 5) reach ratios close to or >1.0, while others remain far below therapeutic thresholds?
  2. How do absolute ECF concentrations (e.g., Patient 2: ~24 mg·h/L vs. Patient 4: ~134 mg·h/L) compare to expected MIC targets for CNS pathogens?
  3. Does plasma AUC24 (range 255–986 mg·h/L) correlate with ECF AUC24 across patients, or are these values largely independent?

Clinical outcome relevance

  1. Did “high penetrators” achieve better infection resolution than “low penetrators”?
  2. Could patients with ratios <0.1 (e.g., Patient 2) be considered at risk for therapeutic failure despite adequate plasma levels?
  3. Should therapeutic drug monitoring (TDM) focus more on ECF exposure than plasma exposure in CNS infections?

Please mention SD in ± instead of brackets ().

In Table 2:

Duplicate time intervals – Patient 3 has two sets of values for “4–8 h” and “8–12 h,” which looks like a reporting or labeling error. Were these meant to be different time blocks?

Inconsistent post-dose ranges – Patients have very different time intervals (e.g., 1–6 h, 7–12 h, 23–30 h). Why was sampling not standardized across patients for comparability?

Unclear ECF values – In some cases (e.g., Patient 2, 12–18 h, ECF = 0.45 mg/L), ECF values seem implausibly low compared to plasma; is this due to assay sensitivity or calibration issues?

 High ECF/plasma ratios – Patient 5 shows a ratio >1.0 (up to 1.56), suggesting higher ECF than plasma concentrations, which is physiologically questionable. Could this result from overcorrection during in vitro recovery calibration?

In Discussion and Conclusion:

The study highlights substantial interindividual variability in vancomycin exposure in brain ECF. The authors are encouraged to explore potential biological and clinical determinants (e.g., BBB integrity, inflammatory status, renal clearance, neurosurgical interventions) that may explain these differences.

In some patients, ECF concentrations appear negligible. The manuscript would benefit from a discussion of whether current dosing regimens are sufficient for this subgroup and whether therapeutic failure is a realistic risk.

The authors may wish to consider and discuss the role of therapeutic drug monitoring (particularly in CSF or brain ECF where feasible) to individualize vancomycin dosing strategies in clinical practice.

A comparative evaluation of vancomycin with other antimicrobials used for post-surgical CNS infections could provide valuable clinical context and help highlight the unique challenges with vancomycin penetration.

The need for further studies is well stated; however, the authors could strengthen their conclusion by suggesting more specific research avenues, such as multicenter prospective studies, development of predictive pharmacokinetic models, or incorporation of pharmacogenomic/biomarker analyses.

Since the ultimate goal is bacterial eradication, it would be useful to assess or at least discuss how ECF concentrations correlate with microbiological clearance and clinical outcomes.

Final Recommendation

The manuscript provides valuable preliminary insights into vancomycin brain penetration using microdialysis. However, major revisions are required before publication, particularly in the Abstract, Introduction, Results interpretation, and Discussion, to make the work more scientifically robust and clinically impactful.

Comments on the Quality of English Language

The english language in the manuscript is very ordinary and must be improved with strong grammatical usage. The author could take the help of english Service Provider or Native speaker to make it more concise and understadable. I highly recommend the revision of text of this manuscript.

Reviewer 2 Report

Comments and Suggestions for Authors

Dear Authors,

I have reviewed the manuscript entitled "Vancomycin Penetration into Brain Extracellular Fluid in Patients with Postoperative Central Nervous System Infections,"

This is a prospective, observational, microdialysis-based pharmacokinetic study evaluating vancomycin concentrations in brain extracellular fluid (ECF). Five patients undergoing neurosurgery with suspected or confirmed CNS infection were included in the study. Two subgroups with significant interindividual variability in ECF vancomycin penetration ('low penetration' and 'high penetration') were identified.

This study is one of the first to directly measure vancomycin penetration into brain ECF in postoperative CNS infections using microdialysis. Addresses an important clinical issue regarding antibiotic dosing in neurosurgical infections.

The authors use microdialysis, a minimally invasive and sensitive technique, to measure free drug concentrations at the site of infection. The article is generally well written, structured, and supported by detailed tables and supplementary figures.

This study has small sample size and, this is the major weakness. While limitations are acknowledged, the discussion should focus on the main topic to avoid distracting from the main message.

Best regards,

Reviewer 3 Report

Comments and Suggestions for Authors

This study examined the study titled "Vancomycin Penetration in Brain Extracellular Fluid in Patients with Post-surgical Central Nervous System Infections. Exploratory study." Considering the data obtained from the research, its contribution to science is clear. However, it is essential to make the necessary corrections and address the questions I have posed below.

Corrections and questions:

1- The surgical conditions before the infection should be stated.
2- Is there information about the patients' BBB status after the surgery (CT scan, etc.)?
3- Could the lack of febrile status have affected the transmission?
4- Could the severity of the infection have been determined by biomarker analysis from the ECF fluid, and could this have created a different situation?
5- Furthermore, could the low severity of the infection in the cases have caused the system to be affected by the situation described in Line 369? Could this be the reason for the difference in transmission?

Reviewer 4 Report

Comments and Suggestions for Authors

Abstract;

  • What do you mean by the immunological assay mentioned in line 27? Briefly explain what it means in this section.

Introduction

  • Brief information could be provided on the incidence of post-surgical CNS infections, including how frequently they occur and which types of neurosurgical procedures are most commonly associated.
  • The impact of maintaining vancomycin trough levels at 15–20 mg/L versus 20–25 mg/L on treatment success or relapse rates could be addressed.
  • When discussing the clinical relevance of CSF concentrations, it would be helpful to briefly reference findings from previous studies.

Materials and Methods

  • The study included a small number of patients, which limits the generalizability of the findings and the strength of statistical analyses.
  • Why were only 24-hour levels evaluated in the study? Couldn't 48-hour brain ECF levels of vancomycin also be analyzed to provide information about longer-term pharmacokinetic variability?

Discussion

  • In the discussion, please discuss the study findings only by comparing them with the literature. Limitations and future directions for the study should be included under the heading "Limitations and future directions."

Conclusion

  • The conclusion should be a brief summary of your study. Please revise this section to emphasize the findings of your study.

References

  • The references you used are not up-to-date enough. Please double-check and provide the recommended literature for the introduction using current sources.

Reviewer 5 Report

Comments and Suggestions for Authors

the paper entitled;”Vancomycin Penetration in Brain Extracellular Fluid in Patients 

with Post-surgical Central Nervous System Infections. Exploratory study” by Skaistė Žukaitienė et al. deals with an assessment of vancomycin penetration into the brain in patients with suspected or confirmed post-surgical CNS infection. 

The drug concentration in the targeted compartment is a pharmacodynamic requisite for successful antimicrobial therapy in this serious clinical setting.

In this study, Vancomycin exposure in the brain demonstrated marked interpatient variability in post-surgical CNS infections, with some patients showing minimal drug penetration.

The paper reports an apparently very limited case study (only 5 patients?). 

The paper is essentially a few cases study about the perpetration of vancomycin in the cerebrospinal district by measuring its concentration after delivery.

However the paper is interesting and reports potentially valuable measurements about the drug management of these serious infections.

Questions arose in us when reading it:

  1. The introduction is very short and doesn’t have illustrations. We suggest adding a diagram to explain at a glance the study design and the experimental procedure applied
  2. assessment of the integrity in the BBB is a very pertinent issue and addressed in the Discussion. We think an introductory paragraph and remarks about this important issue is important.
  3. microdialysis applied in this clinical context should be better explained, illustrated and compared with options like, for instance, direct drug determination in cerebrospinal fluid which is usually performed elsewhere.

So, why is the microfiber method used instead of other options?

This should be supported by analytical method validation or a citation pointing to that matrix specific method validation previously performed. Are there advantages or drawbacks’

  1. many methods to assess vancomycin concentration are available or even commercial. A methodological revision explaining briefly all available methods and the choice performed here should be provided.
  2. analytical method paragraph which should be completed with a figure reporting at least an example of the experimental readout
  3. PK software determinations adopted should be explained and compared with other possible options in the literature or the market

  1. study design should be explained at a glance with a diagram. Ethical committee approval should be provided, informed consent or a substitution for it, provided the critical conditions of patients. Elaborate on patients number and statistical significance. Otherwise the context is that of a case study.
  2. where there only five cases in this study? this number is provided in the abstract
  3. drug delivery method is very important. A separated section describing drug delivery regimen should be provided separated from the current treatment and data collection paragraph
  4. patients have been ascribed to categories like high or low penetrators. What are the data supporting this categorization? Provide patients classification in results by clinical or laboratory findings 
  5. a diagram explaining patient classification would also be important in anticipating Discussion
  6. A strong limitation is due to a lack of data about germ and their susceptibility to the specific drug. Without that all consideration could be considered far fetched at most ..
  7. There is a vast literature about vancomycin and resistance to it. It should be considered in the Introduction, results and discussion.
  8. avoid using inconclusive assessments in conclusions like “appear sufficient” without any quantification. “not fully understood” is another statement to skip conclusions. Otherwise conclusions are in fact “inconclusive” at all. Sticking to the data at hand, what are the actual supported conclusions?
  9. we suggest down stating the conclusive power of this work and report useful observations 

Round 2

Reviewer 5 Report

Comments and Suggestions for Authors

The paper has been considerably improved since its previous version. Most issues have been addressed convincingly. However still coptions to figure 1 and table 1 should be lenthened to explain their contents to the readers. 

Author Response

Comment 1: However still coptions to figure 1 and table 1 should be lenthened to explain their contents to the readers. 

Response 1: Thank you for the suggestion. The captions for Figure 1 and Table 1 have been expanded as recommended. The revised captions now read as follows:

Figure 1. Schematic study design. Patients with suspected or confirmed post-surgical CNS infection receiving vancomycin and implanted with a microdialysis catheter underwent pharmacokinetic sampling after reaching steady state. Brain ECF (microdialysate), plasma, and CSF (if available) were collected over one dosing interval on Days 1 and 2, followed by clinical outcome monitoring. CSF – cerebrospinal fluid; ECF – extracellular fluid. (lines 248–251)

Table 1. Key patient characteristics. Demographic and clinical data of enrolled patients, including age, sex, renal function (creatinine clearance and dynamic changes), neurological status (GCS score), and underlying surgical condition at study inclusion. (lines 364–366)